# SARS-CoV-2 RT-qPCR testing of pooled saliva samples: A case study of 824 asymptomatic individuals and a questionnaire survey in Japan

Junna Oba[1,2], Hiroaki Taniguchi[1,3,4]*, Masae Sato[1], Masaki Takanashi[5], Moe Yokemura[5], Yasunori Sato[6], Hiroshi Nishihara[1,3,4]

1 Keio Cancer Center, Keio University School of Medicine, Shinjuku-ku, Tokyo, Japan, 2 Department of Extended Intelligence for Medicine, The Ishii-Ishibashi Laboratory, Keio University School of Medicine, Shinjuku-ku, Tokyo, Japan, 3 Research and Development Center for Precision Medicine, University of Tsukuba, Innovation Medical Research Institute, Tsukuba-shi, Ibaraki, Japan, 4 Keio University Hospital Clinical and Translational Research Center, Keio University School of Medicine, Shinjuku-ku, Tokyo, Japan, 5 LSI Medience Corporation Central Laboratory Center, Itabashi-ku, Tokyo, Japan, 6 Department of Preventive Medicine and Public Health, Keio University School of Medicine, Shinjuku-ku, Tokyo, Japan

* h-tani@keio.jp

**Data Availability Statement:** All relevant data are within the manuscript and its Supporting Information files.

## Abstract

From the beginning of the COVID-19 pandemic, the demand for diagnostic and screening tests has exceeded supply. Although the proportion of vaccinated people has increased in wealthier countries, breakthrough infections have occurred amid the emergence of new variants. Pooled-sample COVID-19 testing using saliva has been proposed as an efficient, inexpensive, and non-invasive method to allow larger-scale testing, especially in a screening setting. In this study, we aimed to evaluate pooled RT-qPCR saliva testing and to compare the results with individual tests. Employees of Philips Japan, Ltd. were recruited to participate in COVID-19 screening from October to December 2020. Asymptomatic individuals (n = 824) submitted self-collected saliva samples. Samples were tested for the presence of SARS-CoV-2 by RT-qPCR in both 10-sample pools and individual tests. We also surveyed participants regarding their thoughts and behaviors after the PCR screening project. Two of the 824 individuals were positive by RT-qPCR. In the pooled testing, one of these two had no measurable Ct value, but showed an amplification trend at the end of the PCR cycle. Both positive individuals developed cold-like symptoms, but neither required hospitalization. Of the 824 participants, 471 responded to our online questionnaire. Overall, while respondents agreed that PCR screening should be performed regularly, the majority were willing to undergo PCR testing only when it was provided for free or at low cost. In conclusion, pooled testing of saliva samples can support frequent large-scale screening that is rapid, efficient, and inexpensive.

**Funding:** This work was partially supported by the Keio University Global Research Institute (KGRI) Research Projects for New Coronavirus Crisis: Implementation of a Keio Model to Optimize SARS-CoV-2 PCR Tests through Systems Approach (PI: Koichi Matsuo), the Japan Agency for Medical Research and Development (AMED) (PI: Hiroshi Nishihara, Grant Number 20he1422004j0001), Grant-in-Aid for Scientific Research (C) of JSPS KAKENHI (PI: Junna Oba, Grant Number JP21K10334), and the Ministry of Education, Culture, Sports, Science and Technology of Japan (MEXT) for utilization of the university's PCR equipment. The funding agencies had no role in the study design, collection, analysis, or interpretation of data; in the writing of the report; or in the decision to submit the article for publication.

**Competing interests:** The authors have declared that no competing interests exist.

## Introduction

The COVID-19 pandemic has altered our daily lives. Although vaccination programs have progressed in many countries, daily infections and hospitalizations are still high globally [1,2]. Moreover, breakthrough infections have been reported in fully vaccinated individuals, indicating that such people can still contract and transmit the virus [3–5]. One of the challenges in controlling this disease is that the spread of SARS-CoV-2 infection occurs not only in symptomatic patients but also in asymptomatic carriers, including many who later develop symptoms (presymptomatic cases) [6–8]. Further, as community transmission continues, new variants are likely to emerge; when we drafted this manuscript, the highly transmissible Delta variant had replaced the majority of strains worldwide, but a newer variant Omicron has started replacing the Delta variant lately [9].

Early identification and isolation of infected individuals through comprehensive screening is effective in minimizing the spread of infection. The gold standard diagnostic test for SARS-CoV-2 infection is real-time reverse transcription–polymerase chain reaction (RT-qPCR) using upper respiratory tract specimens [10]. While nasopharyngeal swabs (NPS) have been the most widely used sample type, collecting NPS samples requires trained personnel and personal protective equipment (PPE), increasing infection risks to both patients and healthcare workers; moreover, it causes discomfort and a potential risk of complications to patients because of the anatomical structures of the nasal cavity [11–13]. These factors have limited the widespread use of NPSs.

Using saliva samples is an attractive alternative. It has advantages over NPS samples: it is much less invasive and allows self-collection, making it more feasible for repeated, frequent testing; it minimizes the infection risk during sample collection; and has lower personnel and PPE costs [14]. Some studies that compared the sensitivity and specificity of SARS-CoV-2 diagnostic RT-qPCR tests between saliva and NPS samples have shown that sensitivity is higher using NPS [15–17], while others show that saliva samples have a higher [14,18,19] or similar [20–22] sensitivity. Recent meta-analyses have shown that there is little difference in sensitivity between NPS and saliva samples for the detection of SARS-CoV-2 nucleic acid [23–25]. Although multiple studies have reported the presence of viral RNA in saliva samples from both symptomatic and asymptomatic patients [21,26,27], the timing of sample collection seems to be important, because the viral load of SARS-CoV-2 in saliva declines after disease onset [18,28,29], highlighting the importance of collecting samples during the early phase of disease. Thus, saliva sampling is not only easier, safer, and less expensive, but also a reliable option for COVID-19 testing.

Pooled testing, in which samples from multiple individuals are combined into a single test, has been shown to be effective in multiple infectious disease screening settings, including for syphilis and HIV [30,31]. During the current COVID-19 pandemic, a pooled-sample testing approach has been reported to save both cost and time when implemented on larger scales [30,32,33]. For pooled testing to be effective, certain baseline parameters should be considered: the prevalence or positivity rate within the community, sensitivity and specificity of the test, and the limits of detection. Multiple studies have published models and algorithms to calculate optimal pool sizes that depend on prevalence and cost reductions relative to individual tests [34–38]. Overall, sample pooling is most advantageous for populations with low prevalence, remaining more time- and cost-efficient than individual testing in populations with positivity rates up to around 30%; however, it offers no benefit when the positivity rate (prevalence) becomes higher [34–38].

In our cohort of healthy and asymptomatic individuals who were scheduled for medical check-ups at the Center for Preventive Medicine at Keio University Hospital, only 3 of 2,342 (0.19%) individuals tested between August and December 2020 had positive results [39], indicating low prevalence among asymptomatic individuals during this period. Taken together,

pooled testing of saliva samples, especially for screening in communities with relatively low prevalence, should allow more efficient utilization of resources and more rapid screening of a greater number of people with faster turnaround times.

We proposed the concept of social PCR testing, which allows safe social and economic activities by routine testing of saliva samples from asymptomatic individuals [39]. Parallel implementation of mass screening coupled with sample pooling has been reported to contribute to the success of COVID-19 control [40–42]. Here, we report the results of SARS-CoV-2 PCR screening tests performed on 824 employees of Philips Japan, Ltd., who had neither symptoms nor close contact with infected individuals during the period of October through December 2020. Further, we analyzed the results of web questionnaires completed by 471 employees who had participated in the PCR screening regarding their attitudes toward mass-screening policies. While numerous studies have evaluated the effectiveness and feasibility of pooled testing for SARS-CoV-2, few have assessed participants' attitudes and perceptions towards pooled testing [43]. We believe the survey results from this study will be informative to policy makers and health care professionals in developing of modifying surveillance and screening programs.

## Materials and methods

### Pooled and individual testing of unknown samples from 824 individuals

Between October and December 2020, employees of Philips Japan, Ltd. were asked to participate in RT-qPCR screening. A total of 824 volunteers who provided informed consent were enrolled in this study (information and consent forms are provided in S1 File). Eligible individuals included those who did not report fever above 37.5˚ C or coughing, and had not been in close contact with COVID-19 patients during the two weeks preceding the test. Participants were instructed to collect 1–2 ml of saliva using a FastGene™ Saliva Collection Kit (Nippon Genetics Co., Ltd., Tokyo, Japan), at home, without restrictions on food timing or intake. Samples were collected at each branch office, kept at 4˚C, and sent to the LSI Medience Corporation Central Laboratory (Itabashi-ku, Tokyo, Japan), where all PCR testing was performed within 24 h of receipt. For both test modes, samples were vortexed for 5 s and centrifuged at 2500 rpm for 3 min. For pooled testing, equal volumes of saliva supernatant (20 μl each) from 10 subjects were pooled. When the number of samples in a pool was less than 10, phosphate buffered saline was added to bring the volume to 200 μl. Each individual or pooled 200-ul sample was mixed with 200 μl of lysis buffer and 20 μl of proteinase K solution for inactivation, followed by vortexing, heating for 10 min at 56˚C, and brief centrifugation. RNA was extracted using the Maxwell RSC 48 Instrument and Viral Total Nucleic Acid Purification Kit (Promega, Madison, WI, USA) and eluted into a total volume of 50 μl. For RT-qPCR, 10 μl of RNA template was mixed with 40 μl of PCR Master Mix and amplified using a SARS-CoV-2 Detection Kit -Multi- (TOYOBO Co., Ltd., Osaka, Japan) targeting the N1 and N2 genes, as well as an IC. This detection kit was developed for research use, but was approved for clinical use by the Japanese government in August 2020. RT-qPCR was performed using the COBAS z480 instrument (Roche Diagnostics K.K., Tokyo, Japan). The kit protocol was followed: 42˚C for 5 min for RT; pre-denaturation at 95˚C for 10 s; and 45 cycles of 95˚C for 5 s and 60˚C for 30 s for amplification. Ct values were determined for N1, N2, and IC. Samples were considered positive for SARS-CoV-2 when either the N1 or N2 target was detected with a Ct < 40.

### Online survey

The 824 subjects who participated in the PCR screening were asked to respond anonymously to an online survey on how they felt about the social PCR screening between May 19 and May

26, 2021. Participants received a clear explanation of the survey procedure and could interrupt or terminate the survey at any time without giving a reason.

### Statistical analysis

For the questionnaire, sex differences in demographic characteristics and responses were assessed using Pearson's chi-squared test or Fisher's exact test, where appropriate. Statistical significance was set at $p < 0.05$. Data visualization and statistical analyses were performed using R v4.1.1. For visualizing proportions of online survey participants in Japan, an open-source R package 'NipponMap' (https://CRAN.R-project.org/package=NipponMap) was employed.

### Ethics statement

This study was approved by the institutional review board of Keio University (approval number 20200291). The study was conducted according to the principles of the Declaration of Helsinki of 1964 and later versions. Written informed consent was obtained from all individuals who participated in the PCR screening tests, and informed consent was obtained from all individuals who participated in the survey, all prior to data collection.

## Results

### Pooled versus individual testing of 1060 samples

Multiple studies have shown that neither the presence, absence, nor the severity of symptoms are necessarily associated with the Ct values [21,26,27,44,45]. This finding indicates that asymptomatic and/or presymptomatic individuals can be as infective as symptomatic patients. In general, highly-transmissible individuals tend to have high viral loads (low Ct values). Our focus was to detect highly-transmissible individuals without suspicious symptoms to mitigate further spread of the virus through the use of pooled testing in a screening setting [39]. Our preliminary results of pooled testing using known positive and negative samples are shown in Supporting Information (S2 File, S1 Table).

Saliva samples from 824 subjects without symptoms or known exposure in the preceding two weeks were tested in 10-sample pools and individually. Of the 824 individuals, 212 submitted two samples at different times during the study period; six submitted three times, one submitted four times, one submitted five times, and one submitted six times. The remaining 603 submitted samples once. Two of the 1060 samples (two of the 824 individuals) were positive. In case 1, 10-sample pool testing detected the N1 signal (Ct = 33.71) but did not detect N2, while individual testing detected both N1 and N2 (Ct 33.84 and 35.82, respectively) (Fig 1A, Table 1). In case 2, neither N1 nor N2 signals were detected in pooled testing, whereas individual testing identified both N1 and N2 (Ct 31.81 and 33.24, respectively) (Fig 1B, Table 2). In both cases, individual testing showed sigmoid amplification curves for N1 and N2 at Ct values between 30 and 35. In case 2, the 10-sample pool exhibited no Ct value, but showed a line of amplification towards the end of the cycle (red dotted box in Fig 1B). However, neither of the cases had Ct values or showed any late amplification of N2 in pools.

### Screening summary and PCR amplification inhibition

Overall, for 1060 samples, 110 pooled- and 1060 individual tests were performed (Table 3). Only one pool was scored as positive; individual tests confirmed one positive case from this pool. PCR amplification was sometimes inhibited, as indicated by increased Ct for the IC genes. In general, the IC Ct values in most samples were 20–30. We defined inhibition when

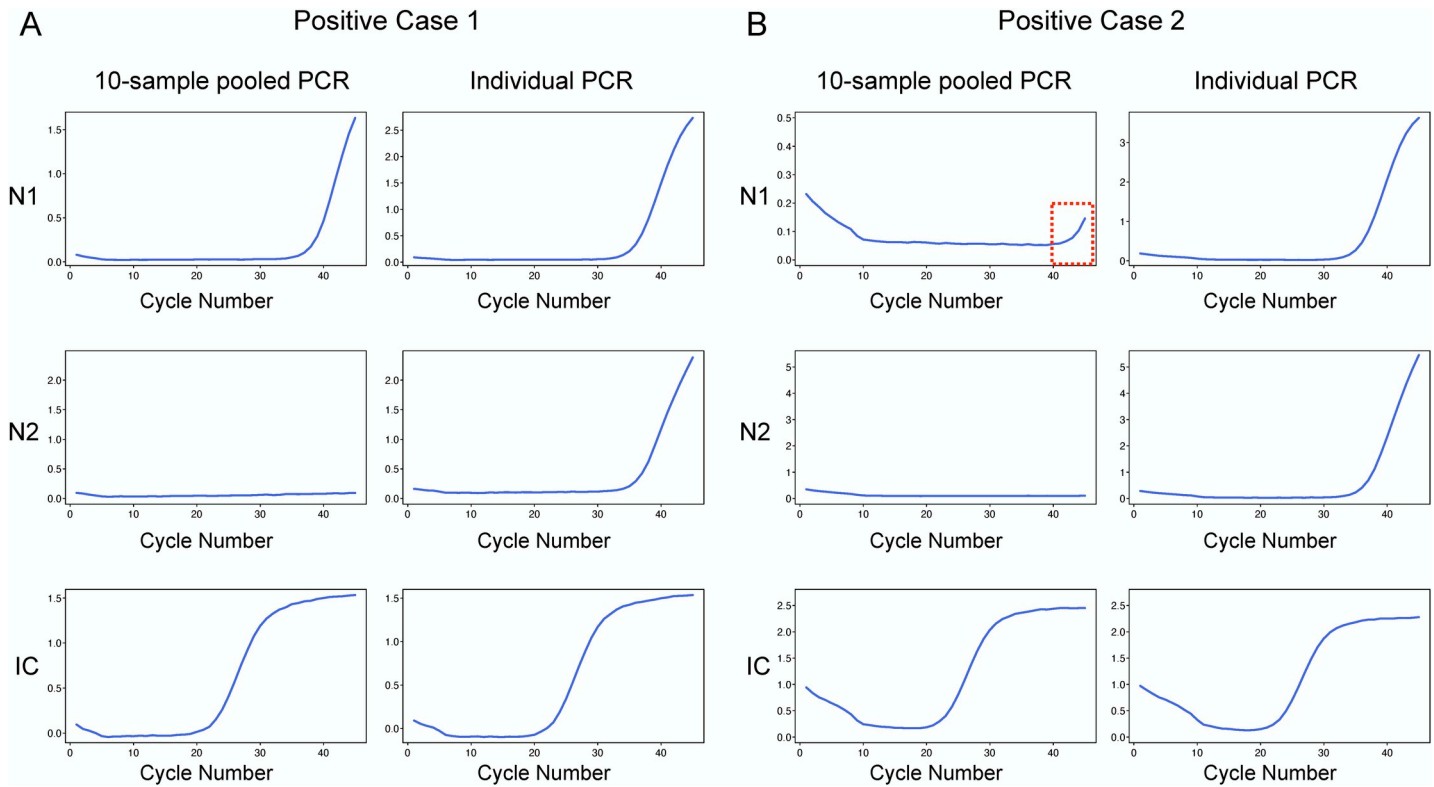

**Fig 1.** PCR amplification curves for positive case 1 (A) and positive case 2 (B). Upper panel, N1; middle panel, N2; lower panel, internal control (IC) gene. Left column of each panel, 10-sample pools; right column of each panel, individual samples.

an IC Ct was unmeasurable or above 40, or when we observed no sigmoid amplification pattern (Fig 2). In total, 74 individual samples and one pooled sample were in this category (Table 3). Interestingly, these 74 samples showed normal range IC Ct values in pooled testing. These results indicate that pooling may have mitigated the inhibition of amplification observed in individual tests.

**Table 1. Individual amplification of 10 samples from the pool including positive case 1 (#8).**

| Sample Name | Result | | Ct (N1) | | Ct (N2) | | Ct (IC) | |
|---|---|---|---|---|---|---|---|---|
| | Individual | Pool | Individual | Pool | Individual | Pool | Individual | Pool |
| #1 | Not detected | Positive | - | 33.71 | - | - | 22.76 | 22.81 |
| #2 | Not detected | | - | | - | | 22.88 | |
| #3 | Not detected | | - | | - | | 22.80 | |
| #4 | Not detected | | - | | - | | 22.72 | |
| #5 | Not detected | | - | | - | | 22.85 | |
| #6 | Not detected | | - | | - | | 23.13 | |
| #7 | Not detected | | - | | - | | 22.73 | |
| #8 | Positive | | 33.84 | | 35.82 | | 22.84 | |
| #9 | Not detected | | - | | - | | 22.80 | |
| #10 | Not detected | | - | | - | | 22.73 | |

When the Ct value for either N1 or N2 was <40, samples were scored as positive; for unmeasurable Ct values shown as dashes, the both test modes were scored as not detected.

**Table 2. Individual amplification of 10 samples from the pool including positive case 2 (#20).**

| Sample Name | Result | | Ct (N1) | | Ct (N2) | | Ct (IC) | |
|---|---|---|---|---|---|---|---|---|
| | Individual | Pool | Individual | Pool | Individual | Pool | Individual | Pool |
| #11 | Not detected | Not detected | - | - | - | - | 26.58 | 29.67 |
| #12 | Not detected | | - | | - | | 28.31 | |
| #13 | Not detected | | - | | - | | 26.94 | |
| #14 | Not detected | | - | | - | | 27.10 | |
| #15 | Not detected | | - | | - | | 29.88 | |
| #16 | Not detected | | - | | - | | 28.05 | |
| #17 | Not detected | | - | | - | | 28.68 | |
| #18 | Not detected | | - | | - | | 32.97 | |
| #19 | Not detected | | - | | - | | 27.26 | |
| #20 | Positive | | 31.81 | | 33.24 | | 28.67 | |

Samples were scored as positive when either N1 or N2 was detected with a Ct value of less than 40; samples were scored as not detected when Ct values were unmeasurable, shown as dashes.

## Clinical courses of the two positive cases

Case one felt chills six days prior to sample collection and visited the local clinic the next day, presenting symptoms including nasal discharge, sore throat, and sputum. She was diagnosed as having a common cold, not tested for SARS-CoV-2, and was prescribed medicine. She still had cold-like symptoms without fever when she submitted her sample. On receiving the positive PCR test result, she underwent another PCR test with a saliva sample at a clinic two days after our screening, which was negative. She had no history of travel or exposure. She self-quarantined and monitored her symptoms for two weeks. She had mild cold-like symptoms for 9–10 days, but developed no fever and required no treatment.

Case two had no symptom when he submitted his sample. However, he started to cough the next evening when he received the positive test result, and developed more serious symptoms in the next morning, including fatigue, chills, fever (38.9°C), headache, joint pain, sore throat, and severe coughing. He tested positive at a local clinic two days after the initial test. When questioned regarding possible exposure, he revealed that he had attended a wedding reception

**Table 3. Pooled and individual testing.**

| Date of sample receipt | Number of samples tested | 10-sample pool PCR | | | | Individual PCR | | |
|---|---|---|---|---|---|---|---|---|
| | | Number of pools | Positive | Not detected | Amplification inhibition | Positive | Not detected | Amplification inhibition |
| 2020/10/22 | 87 | 9 | 0 | 9 | 0 | 0 | 87 | 0 |
| 2020/11/05 | 101 | 11 | 1 | 10 | 0 | 1 | 97 | 3 |
| 2020/11/13 | 272 | 28 | 0 | 28 | 0 | 0 | 268 | 4 |
| 2020/11/19 | 41 | 5 | 0 | 4* | 1 | 1 | 39 | 1 |
| 2020/11/26 | 88 | 9 | 0 | 9 | 0 | 0 | 87 | 1 |
| 2020/12/03 | 137 | 14 | 0 | 14 | 0 | 0 | 136 | 1 |
| 2020/12/10 | 130 | 13 | 0 | 13 | 0 | 0 | 117 | 13 |
| 2020/12/11 | 70 | 7 | 0 | 7 | 0 | 0 | 60 | 10 |
| 2020/12/17 | 134 | 14 | 0 | 14 | 0 | 0 | 93 | 41 |
| Total | 1060 | 110 | 1 | 108 | 1 | 2 | 984 | 74 |

*There was a sharp amplification curve for N1 at Ct higher than 40 in one pool.

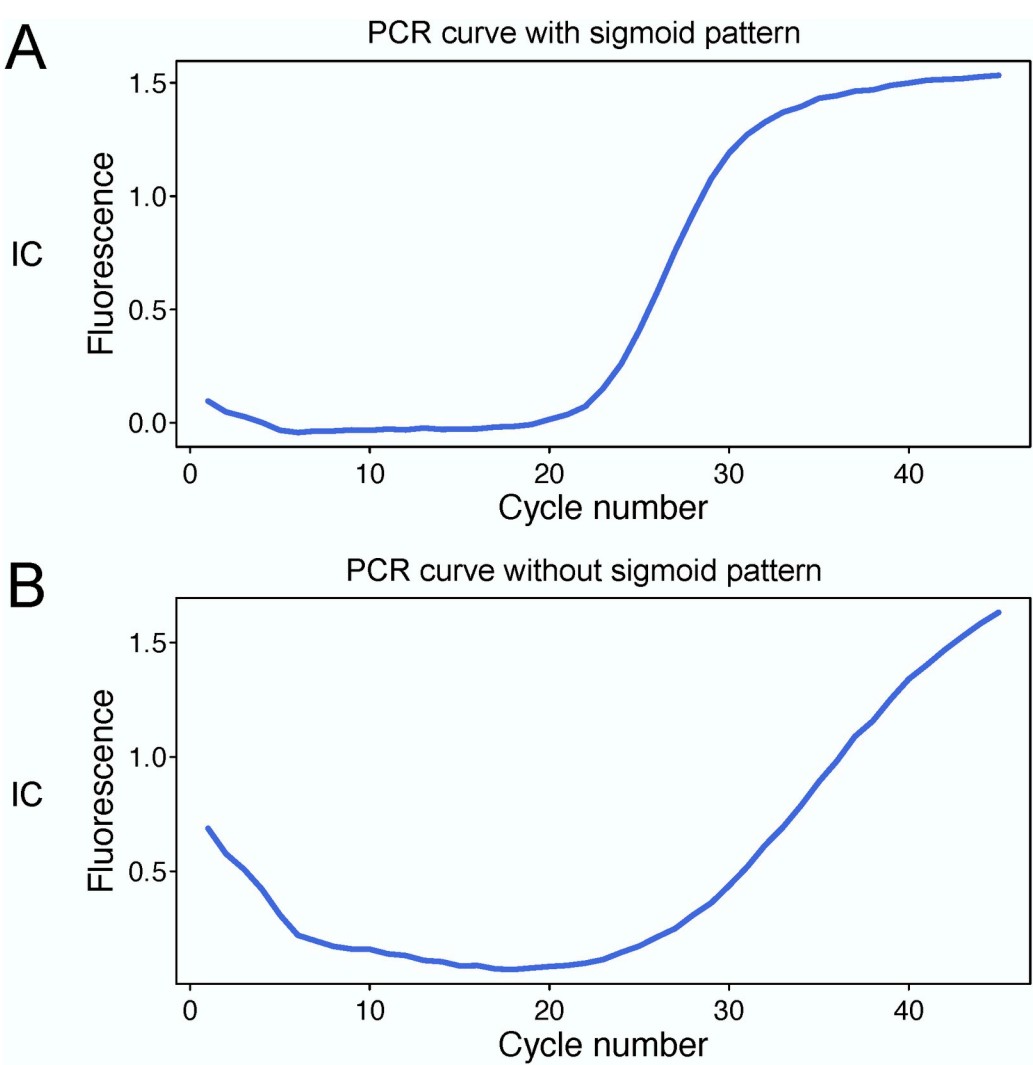

**Fig 2. Examples of good and poor PCR amplification.** (A) Good amplification with a sigmoid pattern. (B) Poor amplification lacking a sigmoid pattern.

three days prior to the initial sampling, and took off his mask while eating the meal. The fever subsided within two days, but symptoms of fatigue, coughing, sore throat, diarrhea, and altered smell and taste continued for 6–7 days. These symptoms gradually resolved and he required no medical intervention during two weeks of home quarantine.

## Questionnaire

Of the 824 individuals who participated in PCR screening test, 476 (57.8%) completed the online questionnaire. As five respondents indicated that they did not want their survey results to be used for research purposes, we analyzed the responses from the remaining 471. The respondents included 358 males and 113 females, the majority between the ages of 30 and 50 years (Fig 3A). The locations of the participants were distributed across Japan, with the top three sites in the Kanto area (Tokyo, Kanagawa, and Saitama) (Fig 3B).

Although no state of emergency was declared by the government during the study period, 62.8% of the respondents did not travel in the two weeks before testing (Q1, Fig 4). Most

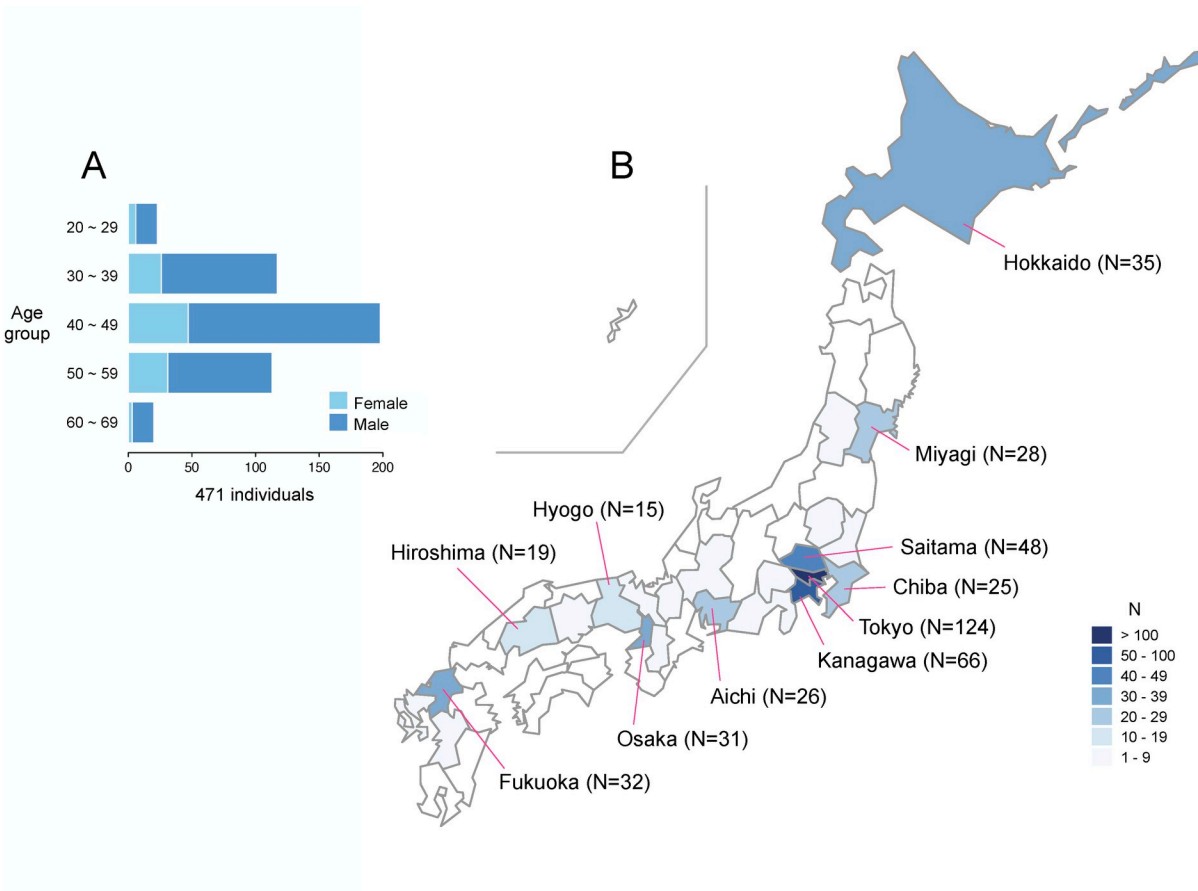

**Fig 3. Characteristics of survey participants.** (A) Age group distribution with males (blue) and females (light blue). (B) Locations at the time of screening, showing prefectures with more than 10 participants. Plots were generated using the survey result (S2 Table), R (version 4.1.1) and an open-source R package 'NipponMap' (https://CRAN.R-project.org/package=NipponMap).

(79.8%) reported that they felt relieved by seeing the result (Q2). The 132 who responded "yes" to Q4 (Has your awareness been changed after participating in this screening project?) (N = 132) were subsequently asked to provide more details, and 131 individuals gave analyzable comments: 81 commented on their awareness of infection prevention, stating that they became more careful or felt assured by their preventative measures (washing hands, avoiding crowds, limiting unnecessary outings, etc.); 42 commented on PCR testing: 23 noted that they realized that saliva PCR testing was easy and helpful; and 19 stated that the PCR testing should be performed regularly. The remaining eight commented that the relief of a negative test result is temporary and that PCR is not 100% accurate (Fig 4).

Individuals who answered "yes" to Q5 (Have you, your family member, or someone close contracted COVID-19 after the screening program?) (N = 27) subsequently gave more specifics on who had contracted COVID-19: one reported his/her own infection, four reported that family members or live-in partners had become infected, and four reported that relatives who did not live with them later tested positive, six reported friends, and twelve reported colleagues or acquaintances at their workplace (Fig 4). Those who responded "yes" to Q6 (Do you think that routine PCR screening should be put into practice at the workplace?) were subsequently asked how much they would be willing to pay for such testing. It is notable that while the overwhelming majority of respondents agreed that routine PCR screening should be put into

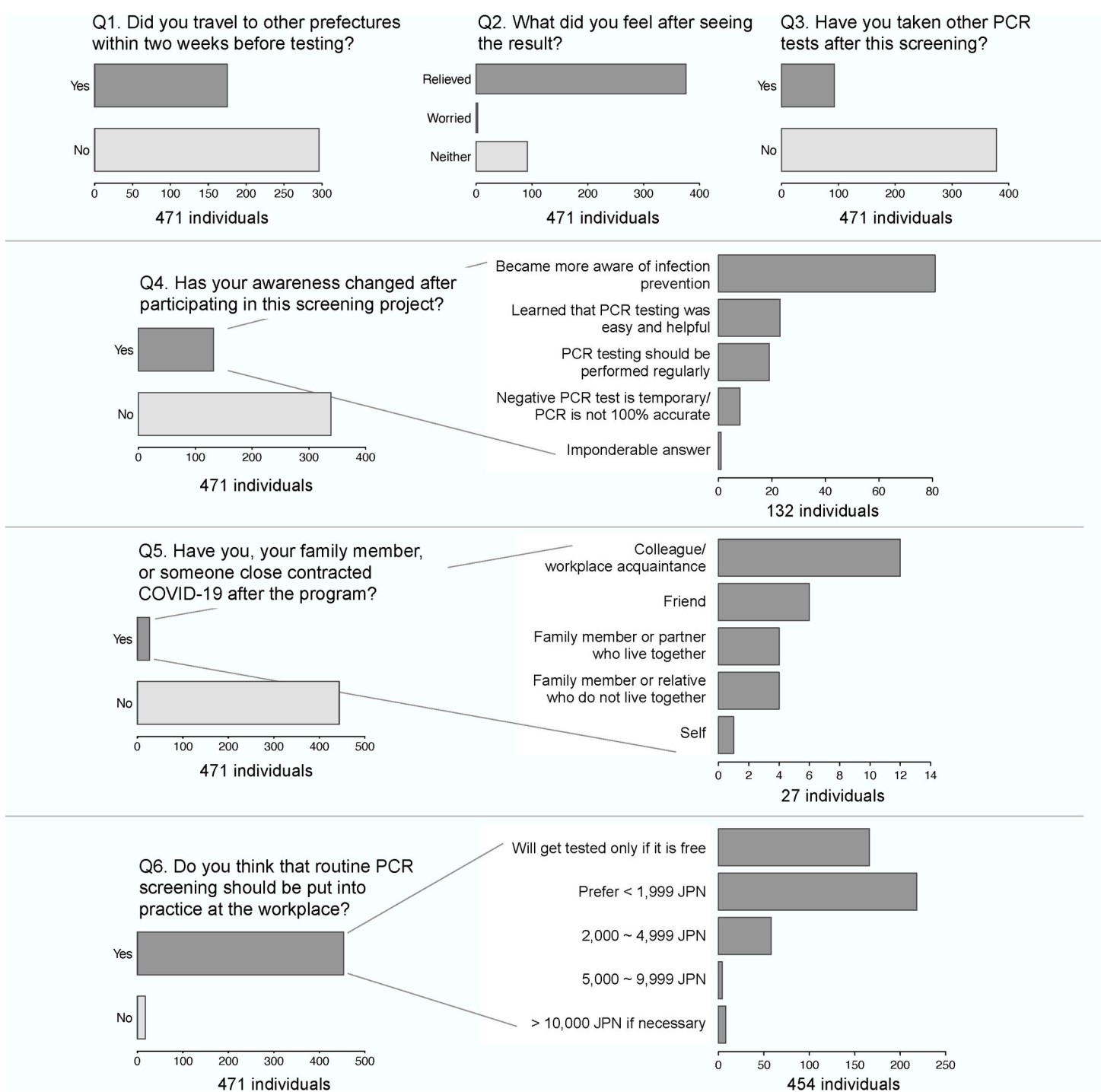

**Fig 4. Responses to the online survey.** Plots were generated using the survey result (S2 Table).

practice at the workplace, many were unwilling to pay out-of-pocket costs; 166 (38.6%) answered that they would not be tested unless the workplace tests were free, and 218 (48%) answered that they would pay if the cost was less than 1,999 JPN (approximately US$16) (Fig 4). Demographic characteristics of the responders, questionnaire responses, and statistical tests for gender differences are summarized in S2 Table. Statistically significant difference

between sexes were seen only for location (Fisher's exact test, p = 0.0005), Q3 (if participants underwent other PCR tests after this program, Chi-squared test p = 0.0078), and Q4-1 (specificity of awareness change, Fisher's exact test p = 0.036).

## Discussion

Although SARS-CoV-2 vaccines reduce the risk of symptomatic and severe disease and infection [46,47], breakthrough infections occur [3–5] and vaccine protective effects decline considerably after six months, especially in older people, according to studies in the UK [48] and Israel [49]. The recent surge of infections with delta and omicron variants across the world demonstrates that vaccine efficacy decreases over time and with immune escape acquired by viral mutations [50–53]. It is important to keep in mind that vaccines alone cannot end the pandemic; therefore, we need to orchestrate all available tools to win the fight against this global scourge, including rapid testing and screening, physical distancing, and sanitary precautions [54].

In this study, we compared the RT-qPCR results of pooled and individual saliva samples from 824 subjects, and reported a follow-up questionnaire survey of 471 of them. Of the 1060 total samples, only those from two individuals were positive (2 of 824 individuals, 0.24%). The positive rate in this cohort was similar to that observed in another cohort of healthy/asymptomatic individuals who underwent RT-qPCR screening before medical checkups at Keio University Hospital during the same period [39]. To compare and contrast our data with official public health surveillance data, we analyzed 1) total numbers of PCR tests and positive cases for suspicious individuals/already diagnosed patients during the same period from the Japan Ministry of Health, Labour and Welfare (https://covid19.mhlw.go.jp/en/) (S3 Table); and 2) monitoring initiatives targeting asymptomatic individuals in 14 prefectures (Hokkaido, Miyagi, Tochigi, Saitama, Chiba, Tokyo, Kanagawa, Aichi, Gifu, Kyoto, Osaka, Hyogo, Fukuoka, and Okinawa) run by the Office for Novel Coronavirus Disease Control, Cabinet Secretariat, Government of Japan (https://corona.go.jp/monitoring/) (S4 Table) in early 2021. In the first case (S3 Table), the mean positivity [(daily cases/tests) × 100] during the period was 13.3%, much higher than that in our cohort. This is probably because of the difference in the target populations; in the first case, PCR tests were performed for suspected individuals with symptoms or recent exposure or contact with confirmed COVID-19 patients. In the second (S4 Table), we see regional and time frame heterogeneity in positive rates, but rates ranged from 0 to 1.34%, comparable to our results.

Because pooling samples will dilute the viral load of a positive sample and tend to produce higher Ct, detecting samples with lower viral loads has been a major challenge in pooling approaches. In our study, case two was not detected in a 10-sample pool. Consistent with our findings, previous studies showed similar Ct increases in pooled testing relative to individual testing, with variations in added values in pooled testing [15,34,55,56]. Interestingly, some groups observed no Ct differences between pooled and individual RT-qPCR testing [57,58]. More et al. reported that while individual positive samples with high viral load (Ct < 28) were consistently detected in pools of 5 or 10, there was a higher frequency of false negatives when samples with lower viral loads (Ct > 28) were pooled, especially in pools of 10. They showed that samples with individual Ct > 31 were not detected in pools of 10, whereas Ct values up to 33 could be detected in a pool of 5; they concluded that pooling up to five samples is more reliable for diagnostic purposes [59]. Praharaj et al. compared 5- and 10-sample pooling and showed that the former had higher concordance with individual testing and lower false-negative rates than the latter; they also showed that 10-sample pools had lower concordance with individual-sample testing, and higher false-negative rates at Ct > 30 [60]. Furthermore,

Watkins et al. reported that sensitivity decreased with increasing pool size: pools of 5, 10, and 20 had reductions of 7.4, 11.1, and 14.8%, respectively [34]. In both of the positive cases in our study, Ct values in individual testing were higher than 30, and 10-sample pools failed to produce any Ct values in case two. Given the decrease in sensitivity in 10-sample pools, and with the two individual positive cases having high Ct values, it is probable that this case would have been undetected if we had not performed simultaneous individual testing.

Mohanty et al. showed that considering late amplification while interpreting the results of pooled samples allowed the identification of additional positives [61]. They first used criteria for positivity (Ct within 35 with a sigmoid curve) for 4-sample pools, but also included an additional class of 'probably positive' (Ct > 35 with non-sigmoid amplification curve, or increased amplification at the end of the reaction). Adding this lenient cut-off yielded 15.5% more true-positive samples. Their study highlighted the importance of catching late amplification to avoid missing positive samples. When carefully reviewing the PCR curves from pooled testing for case two, we observed late amplification patterns at the end of the reaction for N1, as described by Mohanty et al. [61].

We also encountered a relatively high frequency of PCR inhibition in individual tests, defined as Ct values of IC being unmeasurable or above 40, or by no sigmoid PCR amplification pattern. Saliva can contain inhibitors that impair nucleic acid amplification [62,63]. In addition to its molecular composition hindering RNA detection, saliva can be a challenging clinical sample because it varies across individuals in pH and viscosity, as well as being more susceptible to the effects of RNases [62]. Several protocols have been proposed to overcome these challenges, such as dilution, chemical pretreatment, heating, and treatment with proteinase K [63]. In our study, pooled sample testing exhibited less amplification inhibition than individual testing, suggesting that pooling may reduce saliva's inhibitory effects on PCR.

As case one had cold-like symptoms six days prior to sample collection, strictly speaking, she was not asymptomatic. When she underwent another RT-qPCR test at the local clinic, it was eight days after the onset of symptoms, at a relatively late phase of infection. We speculate that this is the reason the test was negative. Case two had no symptoms when he submitted his samples, but had attended a wedding reception three days before. It was very likely that he was infected at the wedding and was presymptomatic (in an incubation period) when he participated in the PCR screening. The RT-qPCR test detected infection before he started to manifest symptoms. This finding reinforces the usefulness and clinical applicability of PCR screening, for example, to detect infection in people who may be dismissed as having common cold or allergy symptoms.

In our study, several participants submitted samples more than once during the study period. There has been increasing evidence that success in containing SARS-CoV-2 depends more on the frequency and turnaround time of the testing than on the test being highly sensitive [64,65].

Multiple factors can account for a high Ct (i.e., a weakly positive case): in very early phases of infection, where viral loads will likely increase, or during recovery, when fragmented viral RNA may be detected but may not be infective. In pooled testing, we may sometimes have negative false results because of decreased sensitivity, especially for samples with low viral loads. To increase the likelihood possibility of detecting vs. missing early infections in cases with high Ct values, repeated, regular testing is recommended in a mass-screening setting [64].

We also collected questionnaire responses from our participants. From the survey result, we could infer that a majority of the participants were taking precaution measures, such as avoiding travel outside of their residing prefectures. The low positive rate of the screening may be because of their precautions, as well as the low prevalence at the time of the study. It is noteworthy that most of the respondents stated that PCR screening should be performed regularly

to ensure a safer work environment. At the same time, the majority were willing to be tested only if screening was free or low cost. The current questionnaire results may be informative to companies, schools, governmental organizations, hospitals, and local communities that are considering screening programs.

In conclusion, pooled RT-qPCR testing of saliva samples is effective and efficient in screening populations with relatively low prevalence. Monitoring a late amplification pattern helps increase detecting positive cases in pooled testing. Regular and frequent screening testing is generally accepted; however, financial costs could be a barrier.

## Supporting information

**S1 Table. Comparison of cycle thresholds between original and pooled SARS-CoV-2-positive and negative samples.** * *For N1, N2, P1, and P2, each sample was first mixed with nine known negative samples, then inactivated with Solution A before PCR. **For P3 and P4, each sample was first inactivated with Solution A, then mixed with other samples before PCR. For unmeasurable Ct values shown as dashes, both test modes were scored as negative.
(XLSX)

**S2 Table. Demographic characteristics of responders, questionnaire responses and sex difference.** a, Fisher's Exact Test; b, Pearson's Chi-squared test with Yates' continuity correction.
(XLSX)

**S3 Table. Daily numbers* of PCR tests, confirmed cases, and positivity rates in Japan, October-December 2020.** *The daily number of PCR tests and confirmed cases in Japan were obtained from https://www.mhlw.go.jp/stf/covid-19/open-data_english.html.
(XLSX)

**S4 Table. PCR test positivity* (%) in 14 prefectures**, March-August 2021.** *Data from https://corona.go.jp/monitoring/. % Positivity = (Number of likely positive cases/Number of PCR tests)*100. **The monitoring initiatives targeting asymptomatic individuals were run by Office for Novel Coronavirus Disease Control, Cabinet Secretariat, Government of Japan.
(XLSX)

**S1 File. Screening information and informed consent forms.**
(DOCX)

**S2 File. Pilot testing with known samples.**
(DOCX)

## Acknowledgments

The authors thank Ms. Kaori Mochida for her technical assistance; Philips Japan, Ltd. for participating in the PCR screening study, conducting the survey, and collecting and sharing the data; Editage (www.editage.com) for English language editing; Dr. Akihito Hagihara for his advice on statistical analysis and interpretation; Dr. Scott E. Woodman (Houston, TX, USA) for proofreading our manuscript, and Dr. A. Gordon Robertson (Courtenay, BC, Canada) for his advice on data visualization, data analysis, and on the manuscript. This research was conducted as part of the Keio Donner Project against COVID-19.

## Author Contributions

**Conceptualization:** Hiroshi Nishihara.

**Data curation:** Junna Oba, Masaki Takanashi, Moe Yokemura.

**Formal analysis:** Junna Oba.

**Funding acquisition:** Junna Oba.

**Investigation:** Junna Oba, Masae Sato, Masaki Takanashi, Moe Yokemura.

**Methodology:** Masae Sato, Masaki Takanashi, Moe Yokemura.

**Supervision:** Hiroaki Taniguchi, Yasunori Sato, Hiroshi Nishihara.

**Visualization:** Junna Oba.

**Writing – original draft:** Junna Oba.

**Writing – review & editing:** Junna Oba, Hiroaki Taniguchi, Yasunori Sato.

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
