## [Decision Letter · Decision Letter 0]

16 Feb 2022

PONE-D-22-02378SARS-CoV-2 RT-qPCR testing of pooled saliva samples: a case study of 824 asymptomatic individuals and a questionnaire survey in JapanPLOS ONE

Dear Dr. Taniguchi,

Thank you for submitting your manuscript to PLOS ONE. After careful consideration, we feel that it has merit but does not fully meet PLOS ONE’s publication criteria as it currently stands. Therefore, we invite you to submit a revised version of the manuscript that addresses the points raised during the review process.

We look forward to receiving your revised manuscript.

Kind regards,

Ruslan Kalendar

Academic Editor

PLOS ONE

Journal Requirements:

"This work was partially supported by the Keio University Global Research Institute

(KGRI) Research Projects for New Coronavirus Crisis: Implementation of a Keio Model to Optimize

SARS-CoV-2 PCR Tests through Systems Approach (PI: Koichi Matsuo), the Japan Agency for Medical

Research and Development (AMED) (PI: Hiroshi Nishihara, Grant Number 20he1422004j0001), Grantin-Aid for Scientific Research (C) of JSPS KAKENHI (PI: Junna Oba, Grant Number JP21K10334), and

the Ministry of Education, Culture, Sports, Science and Technology of Japan (MEXT) for utilization of

the university’s PCR equipment. The funding agencies had no role in the study design, collection,

analysis, or interpretation of data; in the writing of the report; or in the decision to submit the article for publication. "

"This work was partially supported by the Keio University Global Research Institute (KGRI) Research Projects for New Coronavirus Crisis: Implementation of a Keio Model to Optimize SARS-CoV-2 PCR Tests through Systems Approach (PI: Koichi Matsuo), the Japan Agency for Medical Research and Development (AMED) (PI: Hiroshi Nishihara, Grant Number 20he1422004j0001), Grant-in-Aid for Scientific Research (C) of JSPS KAKENHI (PI: Junna Oba, Grant Number JP21K10334), and the Ministry of Education, Culture, Sports, Science and Technology of Japan (MEXT) for utilization of the university’s PCR equipment. The funding agencies had no role in the study design, collection, analysis, or interpretation of data; in the writing of the report; or in the decision to submit the article for publication. "

4. We note that Figure 3 in your submission contain [map/satellite] images which may be copyrighted. All PLOS content is published under the Creative Commons Attribution License (CC BY 4.0), which means that the manuscript, images, and Supporting Information files will be freely available online, and any third party is permitted to access, download, copy, distribute, and use these materials in any way, even commercially, with proper attribution. For these reasons, we cannot publish previously copyrighted maps or satellite images created using proprietary data, such as Google software (Google Maps, Street View, and Earth). For more information, see our copyright guidelines: http://journals.plos.org/plosone/s/licenses-and-copyright.

a. You may seek permission from the original copyright holder of Figure 3 to publish the content specifically under the CC BY 4.0 license.  

5. Please remove your figures from within your manuscript file, leaving only the individual TIFF/EPS image files, uploaded separately.  These will be automatically included in the reviewers’ PDF.

Reviewers' comments:

Reviewer's Responses to Questions

**Comments to the Author**

1. Is the manuscript technically sound, and do the data support the conclusions?

Reviewer #1: Yes

Reviewer #2: Yes

Reviewer #3: Yes

2. Has the statistical analysis been performed appropriately and rigorously? 

Reviewer #1: Yes

Reviewer #2: Yes

Reviewer #3: Yes

3. Have the authors made all data underlying the findings in their manuscript fully available?

Reviewer #1: Yes

Reviewer #2: Yes

Reviewer #3: Yes

4. Is the manuscript presented in an intelligible fashion and written in standard English?

Reviewer #1: No

Reviewer #2: Yes

Reviewer #3: Yes

5. Review Comments to the Author

Reviewer #1: 

The topic the authors have raised is very important, an alternative simple and low-cost approach used for SARS-CoV-2 diagnosis.

Overall, the experimental design was very good, but I have a few concerns:

• A small number of pools were done in the pilot study.

• Theoretically asymptomatic individuals are expected to have a very low viral load. However, the positive samples used for the pilot study have Ct values between low to indeterminate. The study would have benefited from a large number of pools and samples with very high Ct values (like 38, 39) in the pilot study.

• There were only two positive samples from 824 subjects or 1060 samples, and one of them was not detected positive in the pooling experiment, which I think compromised the strength of the work.

• The discussion is unnecessarily too long, it requires significant reduction.

Some additional comments are found in the attached PDF file. The manuscript would have been better if it was page numbered and line numbers given

Reviewer #2: 

The authors present a pilot program of pooled sample saliva testing to monitor for SARS-CoV-2 in a workplace setting, along with a questionnaire to assess the subjects' attitudes regarding testing. The results are useful in that they suggest that the majority of individuals would be willing to submit saliva samples for SARS-CoV-2 screening purposes, especially if the testing was done at no or very little cost to the individual. The authors are appropriately cautious regarding the observed false negative results in their pooled testing, pointing out the existing evidence in the literature regarding 5 sample vs. 10 sample pooling and the possible impact of differences in the precise kit and parameters used for testing. While this is clearly a useful public health tool, I would like the authors to address in their discussion the appropriateness of providing the results of pooled testing directly to individuals given the high risk of a false negative. Should negative test results be provided to individuals given the relatively high risk that those negative results are false and may lead to reduction in compliance with other precautions (masking, distancing, etc) based on the individuals' belief that they are negative for SARS-CoV-2? Should only positive results be reported directly to the individuals to prompt them to undergo further screening? What kind(s) of educational materials should be provided to ensure that participants are fully aware that a positive result is likely to be highly reliable, but a negative result is not?

Reviewer #3: 

The authors report pooled RT-qPCR testing for SARS-CoV-2 from 824 saliva samples obtained from asymptomatic individuals. They also report questionnaire results from a subset of these participants. Overall this is a well-reported study, though one of many in this space. Below are some minor and major comments.

Major:

The introduction would benefit from a rewrite, pointing to any unique aspects of this study over the many other similar studies.

Can you comment on why there is no negative control for pilot experiments (pilot run without P1 -P4)?

Solution A : What is in it? Please elaborate on what is in solution A, and if this is a product please name the company it was purchased from.

"Saliva samples from 824 subjects without symptoms or exposure in the preceding two weeks" should be "known exposure" as exposure to asymptomatic/mildly symptomatic is a very real possibility based on the authors further elaboration on the positive participant.

"Case one was had felt chills six days prior to sample collection and visited the local clinic the next day, presenting symptoms including nasal discharge, sore throat, and sputum. She was diagnosed as having a common cold, not tested for SARS-CoV-2, and was prescribed medicine. She still had cold- like symptoms without fever when she submitted her sample on November 4, 2020. "

- This implies the screening for saliva samples (healthy, without symptoms) was lacking. If the participant had cold-like symptoms when the saliva sample was given she was not asymptomatic, as the authors point out in the discussion. It also implies that reported evasion of symptomatic individuals is not a reliable metric, as clearly SARS-CoV-2 was diagnosed as a minor cold.

The authors point to the minor expense of their method, while obtaining data from the survey about how much workers would be willing to pay for tests. Can the authors elaborate on how suitable this method is based on the workers answers?

Within the discussion it would be preferable if more of a link could be drawn between the saliva screen and the questionnaire results. These seem somewhat separate from each other and further links (in conclusions for example) may improve the discussion.

Minor:

Why not use the same samples between the pilot experiments for direct comparison between both methods (P1 and 2 are missing in the second pilot)?

Table 1: Why 'Probably positive'. It's unclear to me if you are reporting an expectation or a result.

"Living partners", should this be live-in partners?

Participants provided 'informed consent', but the testing was performed at their work place. Can the authors comment on the ethical approval process and address whether workers ay have felt compelled to take part as it was associated with their place of employment?

6. PLOS authors have the option to publish the peer review history of their article (what does this mean?). If published, this will include your full peer review and any attached files.

Reviewer #1: **Yes: **Getachew Tesfaye, Beyene

Reviewer #2: No

Reviewer #3: No

---

## [Author Response · Author response to Decision Letter 0]

25 Mar 2022

Response to Reviewers

Reviewer #1's comments:

The topic the authors have raised is very important, an alternative simple and low-cost approach used for SARS-CoV-2 diagnosis.

Overall, the experimental design was very good, but I have a few concerns:

• A small number of pools were done in the pilot study.

 We are thankful for the reviewer’s positive comment on the overall manuscript and for pointing out concerns regarding our pilot study. We agree that the pilot study had a very small number of pools. In addition, the methods used for the pilot study and the screening program were different. Therefore, we have moved the pilot study part to a supporting information section. 

• Theoretically asymptomatic individuals are expected to have a very low viral load. However, the positive samples used for the pilot study have Ct values between low to indeterminate. The study would have benefited from a large number of pools and samples with very high Ct values (like 38, 39) in the pilot study.

 We are thankful for the reviewer’s comment. We agree with the reviewer that the pilot study would have been much stronger if we had samples with very high Ct values, and if we had tested various pool sizes. In general, highly-transmissible individuals tend to have low PCR Ct values (Ct < 30). Our focus has been that pooled testing in a screening setting can detect highly-transmissible individuals to mitigate further spread of the virus (Oba J et al, Keio J Med 2021). We therefore did not pursue and wait for available positive samples with very high Ct values, and used the samples we could obtain at the time of the study. We would also like to mention that neither the presence, absence, nor the severity of symptoms were associated with the Ct values, as shown in multiple studies (Yang Q et al., Proc Natl Acad Sci U S A 2021; Singanayagam A et al., Euro Surveill 2020; Nikolai LA et al., Int J Infect Dis 2020). This raises awareness that asymptomatic and/or presymptomatic individuals can be as infective as symptomatic patients. However, with the possibility of false negative results from pooled testing, regular testing would be necessary in order to detect individuals with high viral loads, and to mitigate the spread of infection. 

• There were only two positive samples from 824 subjects or 1060 samples, and one of them was not detected positive in the pooling experiment, which I think compromised the strength of the work.

 We appreciate the reviewer’s comment on our finding that only two out of 1060 samples were positive for SARS-CoV-2 by individual RT-qPCR testing and that one of the two positive samples was missed in a pooled testing. We aimed to be transparent and honest with our findings and wanted to propose a cautionary note to readers on interpreting pooled-sample testing results, especially when done alone (without parallel individual testing). Our two positive cases had high Ct values (>30) in individual RT-qPCR, indicating that they both had relatively low viral loads and thus posed a low threat to infect others at the time of sample collection. The strength of pooled-sample testing lies in the ability to save on costs, PPE, and time. Furthermore, it helps to detect a case with a high viral load (with high infectivity to others). As it is possible that a positive case with a low viral load (high PCR Ct) may have a higher viral load later, pooled testing should be done frequently. This ensures that people in a work environment or community can continue their social and economic activities while reducing the risk of viral transmission.

• The discussion is unnecessarily too long, it requires significant reduction.

 We appreciate the reviewer’s comment and have reduced the word count in the discussion section.

• Some additional comments are found in the attached PDF file. The manuscript would have been better if it was page numbered and line numbers given

 We thank the reviewer for paying attention to details and providing helpful comments. We have revised our manuscript as suggested.

Reviewer #2: 

The authors present a pilot program of pooled sample saliva testing to monitor for SARS-CoV-2 in a workplace setting, along with a questionnaire to assess the subjects' attitudes regarding testing. The results are useful in that they suggest that the majority of individuals would be willing to submit saliva samples for SARS-CoV-2 screening purposes, especially if the testing was done at no or very little cost to the individual. The authors are appropriately cautious regarding the observed false negative results in their pooled testing, pointing out the existing evidence in the literature regarding 5 sample vs. 10 sample pooling and the possible impact of differences in the precise kit and parameters used for testing. While this is clearly a useful public health tool, I would like the authors to address in their discussion the appropriateness of providing the results of pooled testing directly to individuals given the high risk of a false negative. Should negative test results be provided to individuals given the relatively high risk that those negative results are false and may lead to reduction in compliance with other precautions (masking, distancing, etc) based on the individuals' belief that they are negative for SARS-CoV-2? Should only positive results be reported directly to the individuals to prompt them to undergo further screening? What kind(s) of educational materials should be provided to ensure that participants are fully aware that a positive result is likely to be highly reliable, but a negative result is not?

 We appreciate the reviewer’s comments on communicating with participants regarding the screening test results, given that there may be false-negative cases from the pooled testing. We provided very clear explanations regarding the screening program. We have clarified that the test was meant to confirm that the tested individual did not secrete enough virus to infect others, but not to exclude the possibility of SARS-CoV-2 infection. We made it clear to the participants that they should continue their daily infection prevention measures. We have provided the information and consent forms translated from Japanese to English as a supporting information. I have copied and pasted statements regarding this matter:

“...Even if your screening test comes back as negative, you can get infected later. Please continue your daily infection prevention measures. 

This test is going to see if virus in saliva is detectable or not, but not exclude the possibility of current or future infection with SARS-CoV-2…”

Reviewer #3: 

The authors report pooled RT-qPCR testing for SARS-CoV-2 from 824 saliva samples obtained from asymptomatic individuals. They also report questionnaire results from a subset of these participants. Overall this is a well-reported study, though one of many in this space. Below are some minor and major comments.

Major:

• The introduction would benefit from a rewrite, pointing to any unique aspects of this study over the many other similar studies.

 We are grateful for the reviewer’s suggestion. We believe that the survey result is very unique to this study and have added such statements in the introduction. 

• Can you comment on why there is no negative control for pilot experiments (pilot run without P1 -P4)?

 We are very thankful for the reviewer’s paying careful attention to our manuscript and data. We did run the negative control for 10-pool sample but did not show its results in the supplementary table in our previous submission. Upon considering the comment by the reviewer, we agree that showing negative control results would be both informative and necessary. We have thus added the results into Supplementary Table S1 (for reference, pasted below).

• Solution A : What is in it? Please elaborate on what is in solution A, and if this is a product please name the company it was purchased from.

 We appreciate the reviewer’s paying attention to such details, and apologize for not having provided clear information on solution A. This solution is included in the SARS-CoV-2 Direct Detection RT-qPCR Kit manufactured by Takara Bio Inc. Although we know that this solution contains elements to inactivate the virus, the precise contents of this reagent are not publicly available. We have clarified that solution A is contained in the kit from Takara Bio Inc. in the supporting information (where we describe our pilot study).

• "Saliva samples from 824 subjects without symptoms or exposure in the preceding two weeks" should be "known exposure" as exposure to asymptomatic/mildly symptomatic is a very real possibility based on the authors further elaboration on the positive participant.

 We would like to thank the reviewer for this suggestion. We agree with this comment and have changed from ‘exposure’ to ‘known exposure,’ since it captures the idea more accurately.

• "Case one had felt chills six days prior to sample collection and visited the local clinic the next day, presenting symptoms including nasal discharge, sore throat, and sputum. She was diagnosed as having a common cold, not tested for SARS-CoV-2, and was prescribed medicine. She still had cold-like symptoms without fever when she submitted her sample on November 4, 2020. "

- This implies the screening for saliva samples (healthy, without symptoms) was lacking. If the participant had cold-like symptoms when the saliva sample was given she was not asymptomatic, as the authors point out in the discussion. It also implies that reported evasion of symptomatic individuals is not a reliable metric, as clearly SARS-CoV-2 was diagnosed as a minor cold.

 We thank the reviewer for making this important point. We agree that in everyday practice people with SARS-CoV-2 infections may have been treated or dismissed as having a ‘mild cold’ without being tested. Our screening program aimed to test individuals without obvious symptoms (fever, and coughing) or a history of close contact with people with confirmed SASR-CoV-2 infection. However, it could potentially include those who had mild symptoms presumably because they don’t consider themselves, or are not considered as being infected with SARS-CoV-2. We can also speculate that the distinction between asymptomatic and presymptomatic individuals can be made only retrospectively. We thus believe that screening projects targeting ‘asymptomatic’ individuals may include presymptomatic individuals and those with mild symptoms who are considered as having a minor cold.

• The authors point to the minor expense of their method, while obtaining data from the survey about how much workers would be willing to pay for tests. Can the authors elaborate on how suitable this method is based on the workers answers?

 We appreciate the reviewer’s comment. From our online survey, we understood that participants preferred to have lower cost testing. Pooled testing saves in the costs for PCR testing, including test kits, reagents, and PPE at medical facilities/laboratories. If sample collection and PCR testing are done at the same place (hospitals or laboratories), we can expect the maximum cost reduction. If self-collected samples are sent from households to laboratories, there will be logistics cost added to the test, reducing the cost reduction benefit from pooled testing. However, even if the logistics cost is included, pooled testing will still be a more inexpensive and plausible option for healthy and asymptomatic individuals, than individual testing.

• Within the discussion it would be preferable if more of a link could be drawn between the saliva screen and the questionnaire results. These seem somewhat separate from each other and further links (in conclusions for example) may improve the discussion.

 We thank the reviewer for the comment. We have added a description of a possible link between the screening and survey results in the discussion.

Minor:

• Why not use the same samples between the pilot experiments for direct comparison between both methods (P1 and 2 are missing in the second pilot)?

 We thank the reviewer for the input. Unfortunately, for the pilot study we had to use samples that remained after clinical use at Keio University Hospital, and P1 and P2 samples were not available for the second pilot study. 

• Table 1: Why 'Probably positive'. It's unclear to me if you are reporting an expectation or a result.

 We thank the reviewer for the helpful suggestion. We have changed from ‘Probably positive’ to ‘positive’ in order to avoid confusing readers.

• "Living partners", should this be live-in partners?

 We are thankful for the reviewer for their careful checking. We have corrected ‘living partners’ to ‘live-in partners.’ 

• Participants provided 'informed consent', but the testing was performed at their work place. Can the authors comment on the ethical approval process and address whether workers may have felt compelled to take part as it was associated with their place of employment?

 We thank the reviewer for the comment. When the PCR screening program was announced, it was made clear that participation in the project was completely voluntary. The decision to or not to participate had no effect on their workplace environment. In addition, the information of who participated and who didn’t was not shared across branches or sectors within the institution. And finally, approximately only 50% of possible employees took part in the pooled testing program. 

Response to editor comments

Journal Requirements:

 Thank you for your suggestions. We have carefully reviewed these two documents and changed the style of our manuscript to meet PLOS ONE's style requirements. 

"This work was partially supported by the Keio University Global Research Institute (KGRI) Research Projects for New Coronavirus Crisis: Implementation of a Keio Model to Optimize SARS-CoV-2 PCR Tests through Systems Approach (PI: Koichi Matsuo), the Japan Agency for Medical Research and Development (AMED) (PI: Hiroshi Nishihara, Grant Number 20he1422004j0001), Grantin-Aid for Scientific Research (C) of JSPS KAKENHI (PI: Junna Oba, Grant Number JP21K10334), and the Ministry of Education, Culture, Sports, Science and Technology of Japan (MEXT) for utilization of the university’s PCR equipment. The funding agencies had no role in the study design, collection, analysis, or interpretation of data; in the writing of the report; or in the decision to submit the article for publication. "

We note that you have provided funding information that is not currently declared in your Funding Statement. However, funding information should not appear in the Acknowledgments section or other areas of your manuscript. We will only publish funding information present in the Funding Statement section of the online submission form. Please remove any funding-related text from the manuscript and let us know how you would like to update your Funding Statement. Currently, your Funding Statement reads as follows: 

"This work was partially supported by the Keio University Global Research Institute (KGRI) Research Projects for New Coronavirus Crisis: Implementation of a Keio Model to Optimize SARS-CoV-2 PCR Tests through Systems Approach (PI: Koichi Matsuo), the Japan Agency for Medical Research and Development (AMED) (PI: Hiroshi Nishihara, Grant Number 20he1422004j0001), Grant-in-Aid for Scientific Research (C) of JSPS KAKENHI (PI: Junna Oba, Grant Number JP21K10334), and the Ministry of Education, Culture, Sports, Science and Technology of Japan (MEXT) for utilization of the university’s PCR equipment. The funding agencies had no role in the study design, collection, analysis, or interpretation of data; in the writing of the report; or in the decision to submit the article for publication. "

 Thank you for your kind suggestion and help. We have removed funding-related text from the manuscript. We request you to use the aforementioned statement (in bold face) instead.

 Thank you for your suggestion. We have validated the corresponding author’s ORCID iD in the Editorial Manager. 

4. We note that Figure 3 in your submission contain [map/satellite] images which may be copyrighted. All PLOS content is published under the Creative Commons Attribution License (CC BY 4.0), which means that the manuscript, images, and Supporting Information files will be freely available online, and any third party is permitted to access, download, copy, distribute, and use these materials in any way, even commercially, with proper attribution. For these reasons, we cannot publish previously copyrighted maps or satellite images created using proprietary data, such as Google software (Google Maps, Street View, and Earth). For more information, see our copyright guidelines: http://journals.plos.org/plosone/s/licenses-and-copyright

a. You may seek permission from the original copyright holder of Figure 3 to publish the content specifically under the CC BY 4.0 license. 

 We thank you for your suggestions. We would like to clarify that Figure 3 was generated by the authors using our own data (Table S2) and R (version 4.1.1) with an open-source R package ‘NipponMap’ (https://CRAN.R-project.org/package=NipponMap). We believe that suggestions made by you do not apply to our Figure 3, however, we have added the description of using the package in the Methods and in the figure legend to ensure the transparency of our work. 

5. Please remove your figures from within your manuscript file, leaving only the individual TIFF/EPS image files, uploaded separately. These will be automatically included in the reviewers’ PDF.

 Thank you for your instruction. We have removed the figures from within our manuscript and uploaded each figure separately as individual TIFF files.

 Thank you for your instruction. We have added captions for Supporting Information files at the end of the manuscript according to your guidelines. 

 Thank you for your instruction. We have carefully reviewed our reference list and are confident that it is complete and correct. Our reference list does not include any retracted papers.

---

## [Editor Report · Decision Letter 1]

22 Apr 2022

SARS-CoV-2 RT-qPCR testing of pooled saliva samples: a case study of 824 asymptomatic individuals and a questionnaire survey in Japan

PONE-D-22-02378R1

Dear Dr. Taniguchi,

We’re pleased to inform you that your manuscript has been judged scientifically suitable for publication and will be formally accepted for publication once it meets all outstanding technical requirements.

Kind regards,

Ruslan Kalendar

Academic Editor

PLOS ONE

---

## [Editor Report · Acceptance letter]

4 May 2022

PONE-D-22-02378R1 

SARS-CoV-2 RT-qPCR testing of pooled saliva samples: a case study of 824 asymptomatic individuals and a questionnaire survey in Japan 

Dear Dr. Taniguchi:

I'm pleased to inform you that your manuscript has been deemed suitable for publication in PLOS ONE. Congratulations! Your manuscript is now with our production department. 

Kind regards, 

on behalf of

Professor Ruslan Kalendar 

Academic Editor

PLOS ONE